# Evaluation of Immunomodulatory Responses and Changed Wound Healing in Type 2 Diabetes—A Study Exploiting Dermal Fibroblasts from Diabetic and Non-Diabetic Human Donors

**DOI:** 10.3390/cells10112931

**Published:** 2021-10-28

**Authors:** Kimberly Nickel, Ursula Wensorra, Horst Wenck, Nils Peters, Harald Genth

**Affiliations:** 1Research Department, Beiersdorf AG, D-20245 Hamburg, Germany; ursula.wensorra@beiersdorf.com (U.W.); horst.wenck@beiersdorf.com (H.W.); nils.peters@beiersdorf.com (N.P.); 2Institute of Toxicology, Hanover Medical School, D-30623 Hanover, Germany; Genth.Harald@mh-hannover.de

**Keywords:** diabetes, skin, glyoxalase I, skin, MIF

## Abstract

The dermis is the connective layer between the epidermis and subcutis and harbours nerve endings, glands, blood vessels, and hair follicles. The most abundant cell type is the fibroblast. Dermal fibroblasts have a versatile portfolio of functions within the dermis that correspond with different types of cells by either direct contact or by autocrine and paracrine signalling. Diabetic skin is characterized by itching, numbness, ulcers, eczema, and other pathophysiological changes. These pathogenic phenotypes have been associated with the effects of the reactive glucose metabolite methylglyoxal (MGO) on dermal cells. In this study, dermal fibroblasts were isolated from diabetic and non-diabetic human donors. Cultured dermal fibroblasts from diabetic donors exhibited reduced insulin-induced glucose uptake and reduced expression of the insulin receptor. This diabetic phenotype persists under cell culture conditions. Secretion of IL-6 was increased in fibroblasts from diabetic donors. Increased secretion of IL-6 and MIF was also observed upon the treatment of dermal fibroblasts with MGO, suggesting that MGO is sufficient for triggering these immunomodulatory responses. Remarkably, MIF treatment resulted in decreased activity of MGO-detoxifying glyoxalase-1. Given that reduced glyoxalase activity results in increased MGO levels, these findings suggested a positive-feedback loop for MGO generation, in which MIF, evoked by MGO, in turn blocks MGO-degrading glyoxalase activity. Finally, secretion of procollagen Type I C-Peptide (PICP), a marker of collagen production, was reduced in fibroblast from diabetic donors. Remarkably, treatment of fibroblasts with either MGO or MIF was sufficient for inducing reduced PICP levels. The observations of this study unravel a signalling network in human dermal fibroblasts with the metabolite MGO being sufficient for inflammation and delayed wound healing, hallmarks of T2D.

## 1. Introduction

The upper skin (epidermis) consists of a multi-layered, keratinizing squamous epithelium. The outer layer, the stratum corneum, forms the topmost protective barrier of the skin and consists of dead tissue (corneocytes). The germ layer, consisting of the stratum basale which is composed of keratinocytes, and the stratum spinosum, provides replenishment for the stratum corneum and renews it about every four weeks [1]. The keratinocytes migrate up into the stratum spinosum, where they gradually keratinize. In this process, they flatten out, lose their nucleus, and become corneocytes [2]. The connective tissue layer is called dermis and links the epidermis with the subcutis. The dermis is rich in collagen fibres ensuring the skin’s particular elasticity [3,4]. Furthermore, blood and lymph vessels, and the so-called skin appendages, such as hair follicles, sebum, and sweat glands, as well as numerous nerve fibres, are embedded in the dermis. Besides fibroblasts, the major cell type, other cell types found in the dermis are immune cells (macrophages, mast cells), stem cells, adipocytes, Schwann cells [5,6]. Dermal fibroblasts execute plenty of functions via direct contact and autocrine as well as paracrine signalling and fulfil an important role within the homeostasis of the skin and during wound healing [7,8,9]. Fibroblasts produce the progenitors of collagen I—procollagen type I [10]. After secretion, procollagen is cleaved by endopeptidases to procollagen propeptides (C-terminal and N--terminal) and tropocollagen that form the triple helix, which the dermis is mostly composed of. Procollagen type I c-peptide (PICP) has a molecular weight of 100 kDa and its production is directly correlated with the production of collagen, and it is therefore used as a marker for collagen production. Moreover, it is used as a marker for osteoblast activity [11].

Diabetes mellitus type II (T2D) is a pandemic metabolic disease characterized by hyperglycemia, chronic inflammation, and insulin resistance [12]. Going untreated or inadequate medication leads to numerous comorbidities such as nephropathy, fatigue, polydipsia, polyphagia, polyuria, neuropathy, and angiopathy [12,13,14]. Eventually, about 70% of T2D patients also develop cutaneous manifestations including itch, dry skin, impaired wound healing, ulcers, or diabetic dermopathy [14,15,16,17,18].

Methylglyoxal (MGO), a highly reactive dicarbonyl compound, is regarded to be involved in the development of the secondary diseases [19]. High levels of MGO are formed when the concentration of its precursor glucose is elevated (e.g., in hyperglycemia). Furthermore, increased levels of MGO have been associated with decreased levels and/or decreased enzymatic activity of the MGO-detoxifying glyoxalase system [19]. Furthermore, MGO reacts in a non-enzymatic glycation reaction with various cell components (e.g., proteins, DNA), resulting in elevated levels of advanced glycation end products (AGEs). AGEs bind to special transmembrane receptors called RAGE [20]. RAGE activation stimulates signalling pathways regulating inflammation, differentiation, and cell death [19]. MGO induces hyperalgesia by modifying NaV1.8 [21]. Furthermore, constant elevated levels of AGEs and MGO damage vasculature, leading to vasculopathy, as well as nerve endings essential for sensation, but also for different neurocutaneous interactions including a variety of physiological and pathophysiological functions, e.g., cell growth, immune response, inflammation, pruritus, and impaired wound healing [22,23,24]. The main AGE formed by MGO is the hydroimidazolone MG-H1 (>90%) [25].

Interleukin-6 (IL-6) is a pro-inflammatory interleukin which is involved in acute stress response by stimulating the production of acute phase response. Its level is elevated in T2D patients [26]. Macrophage migration inhibitory factor (MIF) is also released to induce acute immune response with longer persistence than IL-6. MIF is very versatile, as it is not only a cytokine but also an enzyme (tautomerase activity), an endocrine hormone, and a chaperon-like molecule [27,28,29].

Fibroblasts are critically involved in wound healing, including the breakdown of the fibrin clot, the formation of new collagen structures and extracellular matrix (ECM), and wound contraction. In T2D and aged patients, increased levels of glycated collagen have been reported [30,31,32]. High levels of glycated collagen in the skin changes not only mechanical behaviour of the skin but also cell behaviour. It was found that in vitro cell-induced material contraction was inhibited and, as well, the actin cytoskeletons of the dermal fibroblasts were reorganized [33]. The main type in the human skin is collagen type I. Collagen type I is produced by fibroblasts as procollagen type I, secreted and processed outside the cells. The strands thereby arrange themselves into long, thin fibrils that cross-link to one another [10]. PICP I, a product during collagen production, is established as an indicator of type I collagen synthesis.

The paradigm of this study is that dermal fibroblasts are key players in the skin pathology (in particular inflammation and reduced wound healing) of T2D patients. In the first part of this study, the properties of dermal fibroblasts from diabetic and non-diabetic donors were differential analysed with regard to the secretion of several immunomodulatory factors and PICP. Furthermore, secretion of these immunomodulatory factors and PICP from cultured diabetic and non-diabetic fibroblasts was analysed upon exposure to glucose metabolite MGO, a condition that mimics the in vivo situation in the skin of diabetic individuals. In the second part of this study, MIF treatment of fibroblasts is presented to result in blocked enzymatic activity of glyoxalase I, which might be a causative for the reduced glyoxalase activity observed in diabetic fibroblasts.

## 2. Materials and Methods

### 2.1. Study Design

A cross-sectional study was performed in cooperation with SIT (Skin Investigation and Technology, Hamburg, Germany). The study was approved by the ethics committee Freiburg (feki-Code 015/1630, amendment 1) and conducted according to the principles of the Declaration of Helsinki. All subjects signed a consent form, and the participants were enrolled in November and December 2015. Ten non-diabetic and ten diabetic subjects (type II) were recruited. The diabetics fulfilled an anamnesis questionnaire, whereas the non-diabetic subjects fulfilled an adapted questionnaire. Before the punch biopsies were taken, blood samples from all subjects were taken to confirm their HbA_1c_ value. Three punch biopsies per subject were taken from the inner side of one upper arm (diameter: 6 mm). The blood sampling and the punch biopsies were taken by medical personnel at the Dermatologikum Hamburg. The wound healing was monitored by follow-up visits 1 to 5 days and 10 to 14 days after the punch biopsy.

The biopsies were collected and stored at 4 °C until transfer to Beiersdorf AG, Hamburg, Germany. The in-vitro analyses were performed by Beiersdorf AG.

### 2.2. Individuals

For the diabetic group, 10 men and women aged between 18 and 65 years with a confirmed diabetes diagnosis at least 5 years prior were recruited. Age and sex-matched non-diabetic volunteers were chosen accordingly. Diabetes medication (e.g., insulin or metformin) was approved; however, subjects with high blood pressure, serious illness, or cardiovascular disease were excluded. HbA_1c_ had to be ≥5.85% while non-diabetic volunteers had to be <5.85%. Lesions, scars, or tattoos on the testing area (inner forearm) were exclusion criteria. A summary of demographic data is given in Table 1.

### 2.3. Isolation of Dermal Fibroblasts

Primary fibroblasts were obtained from punch biopsies. The biopsies were incubated in 2.4 U/mL dispase II for 2 h at 37 °C and then separated into epidermis and dermis by hand. Dermal fibroblasts were isolated from the dermis. The dermis was incubated overnight at 37 °C with 312 U/mL collagenase type I (Merck, Darmstadt, Germany) at 37 °C. After incubation, the cell suspension was filtered through a 70 μm filter (Greiner Bio-One, Kremsmünster, Austria). The dermal fibroblasts were grown in low glucose [1 g/L] DMEM (Gibco/Thermo Fisher Scientific, Waltham, MA, USA) supplemented with 10% [*v/v*] fetal bovine serum (FBS, (Gibco/Thermo Fisher Scientific, Waltham, MA, USA) and 1% [*v/v*] penicillin/streptomycin (Gibco/Thermo Fisher Scientific, Waltham, MA, USA) at 37 °C and 7% CO_2_. For treatments, fibroblasts were grown in DMEM containing 2% FBS. All experiments in this study were performed on fibroblast cultures from passage 4 to 9. Working with primary cells has an impact on reproducibility. The experiments were tested multiple times before used with primary cells. Moreover, the results usually present a means of at least 3 tests per donor in order to stabilize the result.

### 2.4. Nuclear Count

For the examination of the proliferation rate of the fibroblasts, the number of nuclei was determined after 24, 48, and 72 h. For this purpose, 1 × 10^3^ cells were seeded in wells of a 96-well plate and fixed at intervals of 24 h with acetone for 15 min, then washed and stained with Hoechst 33342 (Merck, Darmstadt, Germany). The evaluation of the nucleus staining was automated on the Axio Observer Z1 (Zeiss, Jena, Germany).

### 2.5. Detection of Cell Proliferation by BrdU Assay

Cell proliferation of human dermal fibroblasts of diabetic and non-diabetic subjects was quantitated using colorimetric 5-bromo-2′-deoxyuridine (BrdU) assay (Roche, Basel, Switzerland) and was performed according to manufacturer’s protocol. Short, 1 × 10^4^ fibroblasts per well were seeded in 96-well plates at 37 °C for 24 h and then treated for another 24 h. BrdU was added and incorporated during DNA synthesis in replicating (cycling) cells during an incubation time of 24 h. Medium was removed, and cells were fixed and denatured. Anti-BrdU-POD was added. The antibody binding was detected by adding substrate (tetramethyl-benzidine). The reaction produces a colorimetric signal that is detected by 450 nm. The absorbance correlates to the amount of DNA synthesis and with that to the number of proliferating cells.

### 2.6. Reactive Oxygen Species

1 × 10^4^ fibroblasts per well were seeded in 96-well plates at 37 °C for 24 h and then treated for another 24 h. Cells were washed three times with PBS. 10 μM 2′,7′-dichlordihydrofluorescein-diacetat (H_2_DCFDA, Thermo Fisher Scientific, Waltham, MA, USA) diluted in PBS was added to the cells. The plate was incubated for 20 min at 37 °C. During that incubation, H_2_DCFDA was oxidized to 2′,7′-dichlorfluorescein (DCF). The fluorescent signal is detected at 493 nm.

### 2.7. Glucose Uptake Assay

In order to analyse the insulin sensitivity of diabetic fibroblasts, insulin-stimulated glucose uptake was investigated. For this purpose, 2 × 10^4^ cells/well of fibroblasts were seeded in a black 96-well plate with a transparent bottom and cultured overnight at 37 °C. in DMEM with 10% FBS. The next day, the medium was changed to starvation medium (DMEM without glucose and FBS) for 5 h. After 5 h, insulin (Merck, Darmstadt, Germany) stimulation (5 μM in starvation medium) started for 10 min at 37 °C. The control cells were cultured in parallel only in starvation medium. The insulin-stimulated cells and the control cells were then washed with PBS and incubated for 30 min with fluorescently labelled 2-(*N*-(7-nitrobenz-2-oxa-1,3-diazol-4-yl) amino)-2-deoxyglucose (2-NBDG; 100 μg/mL; Thermo Fisher Scientific, Waltham, MA, USA). Control cells treated without 2-NBDG were kept as blank. The cells were washed three times with PBS, 100 μL PBS was added, and the fluorescence was measured at an excitation wavelength of 485 nm and an emission wavelength of 535 nm. The insulin-stimulated glucose uptake was then put into perspective on the unstimulated (only 2-NBDG treated) glucose uptake.

### 2.8. Insulin Receptor Detection

For the determination of surface proteins such as the insulin receptor of the dermal fibroblasts, the FACS (fluorescence-activated cell sorter) was used. For this, the cells were trypsinized (Thermo Fisher Scientific, Waltham, MA, USA) and 2 × 10^5^ cells per batch were transferred to tubes. Subsequently, the cells were washed and incubated for 30 min with the primary antibody (1:1000) in PBS + 1% BSA on ice. Subsequently, the cells were washed three times and incubated for 15 min with the secondary antibody on ice in the dark. After three more washes, cells were picked in PBS and examined at FACS. Isotype controls were used as confirmation of specificity.

### 2.9. Detection of IL-6 in Supernatants of Dermal Fibroblasts

IL6 levels in fibroblast supernatants were detected using HTRF^®^ (Homogeneous Time Resolved Fluorescence) assay from Cisbio (Codolet, France) and were measured according to manufacturer’s protocol. Shortly, IL6 was detected in a sandwich assay format using 2 different specific antibodies, one labelled with Eu^3+^-cryptate (donor) and the second with XL (acceptor). When the labelled antibodies bind to the same antigen, the excitation of the donor with a light source (620 nm, laser or flash lamp) triggers a Fluorescence Resonance Energy. They were transferred (FRET) towards the acceptor, which in turn fluoresces at a specific wavelength (665 nm). Signal intensity is proportional to the number of antigen-antibody complexes formed and therefore to the IL6 concentration. 10 μL of the cell supernatants to be examined were treated with 5 μL of the two antibodies. After 3 h, the supernatants were measured with light at the wavelengths 620 nm and 665 nm and the ratio formed therefrom. The value obtained was then converted into pg/mL using the standard series. When cells were pre-treated with substances, the measurement was made 24 h before DMEM with 2% FBS.

### 2.10. MIF Detection in Supernatants of Dermal Fibroblasts

For the analysis of the MIF concentration in supernatants of dermal fibroblasts, the Duoset ELISA by R & D Systems (Minneapolis, MN, USA) was used according to the manufacturer’s instructions. For this purpose, 1 × 10^4^ fibroblasts were seeded into a 96-well microtiter plate and incubated overnight at 37 °C. One hundred μL of the supernatant were pipetted into a 96-well microtiter plate already prepared with primary antibody and incubated for 2 h at 37 °C. Subsequently, the sample was removed and 100 μL of the secondary antibody was added. After an incubation period of 2 h at 37 °C, the plate was washed three times and 100 μL horse radish peroxidase (HRP) was added. After 30 min at 37 °C, the plate was washed three times, then 100 μL of substrate was added to the HRP and incubated at room temperature for 15 min until a uniform, blue coloration had established. Now, 50 μL of the stop solution was added (samples turned yellow) and absorbance was measured at 450 nm. When cells were pre-treated with substances, this was done 24 h before measurement in DMEM with 2% FBS.

### 2.11. MG-H1 Assay

For the analysis of the methylglyoxal 5-hydro-5-methylimidazolone (MG-H1) concentration in lysates of dermal fibroblasts, a competitive ELISA was used according to the manufacturer’s instructions (Elabscience, Houston, TX, USA). When cells were pre-treated with substances, this was done 24 h before measurement in DMEM with 2% FBS. Cells were lysed by 3 thaw and freeze cycles in PBS. Briefly, 100 μL of lysate was pipetted into an already prepared 96-well microtiter plate and incubated for 90 min at 37 °C. Subsequently, the sample was removed and 100 μL of biotinylated detection antibody was added and incubated for 1 h at 37 °C. The plate was washed three times and 100 μL horse radish peroxidase (HRP) was added. After 30 min at 37 °C, the plate was washed five times, then 90 μL of substrate was added and incubated at room temperature for 15 min until a uniform, blue coloration had established. Now, 50 μL of the stop solution was added (samples turned yellow) and absorbance was measured at 450 nm.

### 2.12. Semi-Quantitative Real-Time PCR

For the gene expression analysis by means of real time PCR, 2 × 10^5^ fibroblasts per well were seeded in 6-well plates and cultivated to confluence. The drugs were added overnight in DMEM with 2% FBS. The next day, the total RNA was isolated using RNeasy Mini (Qiagen, Hilden, Germany) following manufacturer’s instructions. Using the High-Capacity cDNA Transcription Kit (Qiagen, Hilden, Germany), 1 μg of mRNA was transcribed into cDNA by following the manufacturer’s instructions. The relative gene expression analysis was performed on the ABI 7900H Fast Real-Time PCR System (Thermo Fisher Scientific, Waltham, MA, USA) using the TaqMan Gene Expression Master Mix and the TaqMan probes according to the genes to be examined (Thermo Fisher Scientific, Waltham, MA, USA). The gene expression change was determined by the number of cycles at which a Threshold Cycle (Ct) was exceeded. The data thus obtained were normalized to an endogenous control and to each other with the untreated, non-diabetic control. Expression of the glo1 gene was quantified and normalized to gapdh as an endogenous control. The relative gene expression was calculated by the ΔΔCt method of Livak and Schmittgen (2001) [33].

### 2.13. Glyoxalase I Protein Expression

For the analysis of glyoxalase I protein expression, the sandwich ELISA by cloud clone was used according to the manufacturer’s instructions. Cells were lysed by 3 thaw and freeze cycles in PBS. Briefly, 100 µL of a cell lysate whose concentration was priorly determined using bicinchoninic acid assay (BC Assay Uptima, Interchim, Montluçon, France) was pipetted into an already prepared 96-well microtiter plate and incubated for 1 h at 37 °C. Subsequently, the sample was removed and 100 μL of the secondary antibody was added. After an incubation period of 1 h at 37 °C, the plate was washed three times and 100 μL horse radish peroxidase (HRP) was added. After 30 min at 37 °C, the plate was washed five times, then 100 μL substrate of HRP was added and incubated at room temperature for 15 min until a uniform, blue colour had set. Now, 100 μL of the stop solution was added (samples turned yellow) and absorbance was measured at 450 nm.

### 2.14. Glyoxalase I Activity

To determine the glyoxalase I activity, cells were lysed by 3 thaw and freeze cycles in 200 mM Tris buffer (+0.1 Triton X-100 (Sigma-Aldrich Co., LLC., St. Louis, MO, USA)) Subsequently, the total protein concentration was determined by BC assay. Five μg of lysate was used to determine enzyme activity. The substrate (hemithioacetal from 200 mM MGO (Sigma–Aldrich Co., LLC., St. Louis, MO, USA) + 200 mM GSH (Sigma–Aldrich Co., LLC., St. Louis, MO, USA), 1 mM final concentration in the sample) was pre-incubated at 37 °C for 10 min. In 200 mM sodium phosphate buffer (pH 6.6), lysate and substrate were then combined to a total volume of 200 μL. Glyoxalase I-mediated formation of S-d-lactoylglutathione was measured photometrically as a change in absorbance at 240 nm at 37 °C for 10 min. The activity was defined as glyoxalase I unit, which can produce 1 μmol of S-d-lactoylglutathione from the hemithioacetal per minute.

### 2.15. Procollagen Type I C-Peptide ELISA

Procollagen Type I C-Peptide directly correlates with the produced collagen fibrils and was therefore detected using the assay from Takara (Takara Bio, Shiga, Japan). For this purpose, 20 μL of a 1:20 diluted sample and a horse-radish-peroxidase (HRP)-conjugated antibody were pipetted into an already prepared 96-well microtiter plate and incubated for 3 h at 37 °C. After four washes with wash buffer, 100 μL of substrate was added to the plate and incubated at room temperature for 15 min until a uniform, blue colour was obtained. One hundred μL of the stop solution was added (samples turned yellow) and absorbance was measured at 450 nm.

### 2.16. Statistical Analysis

Data were expressed in mean ± SD. For statistical analysis STATISTICA 13.3 (TIBCO Software Inc., Palo Alto, CA, USA) was used. The Kolmogorov–Smirnov test and Shapirov–Wilk test were used for normality testing. The Mann–Whitney U-test (unpaired data) and Wilcoxon-sum-test (paired data) were used as well. For this study, a significance level of at least *p* ≤ 0.05 was considered significant.

## 3. Results

### 3.1. Characteristics of the Donors

A total of 10 diabetic and 10 non-diabetic volunteers were invited to the study (Table 1). Each the diabetic and the healthy group comprised three women and seven men. The average age of the diabetic volunteers was 54 ± 9 years, the age of the non-diabetic subjects was 54 ± 9 years. The HbA_1c_ value of the diabetic patients was 7.06 ± 1.03%, the non-diabetic volunteers had an HbA_1c_ value of 5.44 ± 0.22%. Punch biopsies were taken from the inner forearms.

### 3.2. Reduced Insulin-Induced Glucose Uptake and Reduced Expression of the Insulin Receptor in Dermal Fibroblasts of Diabetic Donors

Dermal fibroblasts were isolated from punch biopsies of diabetic and non-diabetic donors. Cultured diabetic and non-diabetic fibroblasts exhibited comparable kinetics of proliferation, as analysed in terms of an increasing number of stained nuclei and BrdU assay (Appendix A). Dermal fibroblasts have been shown to exhibit increased intracellular levels of reactive oxygen species (ROS) under diabetic conditions [34]. To check if the intracellular levels of ROS were increased in dermal fibroblasts as well, ROS formation was evaluated in terms of the DCF assay. The ROS levels of non-treated fibroblasts from diabetic donors were elevated compared to non-diabetic donors (non-diabetic: 32,146 ± 3391 a.u. vs. diabetic: 37,532 ± 3742; Appendix A). Increased ROS levels did, thus, not coincide with changed proliferation of dermal fibroblasts.

Next, glucose uptake and cell surface expression of the insulin receptor (InsR) were investigated. Upon stimulation with Insulin, non-diabetic fibroblasts responded with increased glucose intake (*p* = 0.028) (Figure 1A). In contrast, the basal glucose uptake of diabetic and non-diabetic fibroblasts was comparable (Figure 1A). This implies that the glucose uptake via GLUT1/3 was not impaired in diabetic fibroblasts. Additionally, the cell surface expression of InsR was analysed by flow cytometry (Figure 1B). Therefore, the cells were co-stained for CD90 and insulin receptor (InsR, Appendix A). InsR was detected on cell surfaces of the fibroblasts of both non-diabetic and diabetic donors. However, fibroblasts from non-diabetic donors tended to exhibit more receptors, albeit not significantly (Figure 1B). This observation suggests a defect insulin receptor trafficking (endocytosis and followed by recycling to plasma membrane) in diabetic fibroblasts [35]. In sum, the diabetic phenotype of diabetic fibroblasts seems to persist under cell culture conditions.

### 3.3. Changed Secretion of IL-6, Macrophage Migration Inhibitory Factor, and Procollagen Type I C-Peptide in Fibroblasts of Diabetic Donors

Diabetes is well established to be associated with chronic inflammation [12]. In this regard, the secretion of several important proinflammatory mediators was analysed in the supernatants of dermal diabetic and non-diabetic fibroblasts. The IL6 level was significantly increased in the supernatants of diabetic vs. non-diabetic fibroblasts (78 ± 32 pg/mL vs. 113 ± 30 pg/mL, *p* < 0.05, Figure 2A). Furthermore, the level of macrophage migration inhibitory factor (MIF), also known as glycosylation-inhibiting factor, tended to be increased in the supernatants of diabetic fibroblasts (non-diabetic: 125 ± 32 pg/mL vs. diabetic: 166 ± 62 pg/mL, *p* = 0.14, Figure 2B). Non-significant differences were found, which can be attributed to high variability in the data obtained using primary fibroblasts.

Numerous diabetic patients suffer from wound healing issues, especially on their feet [36,37]. Wound healing involves the formation of a new extracellular matrix (ECM) [38,39,40]. Since dermal fibroblasts are involved in the formation of the ECM, it was investigated whether there is a difference in the production of the procollagen I C peptide between healthy and diabetic fibroblasts, which is directly correlated with the collagen produced. Interestingly, PICP production in the supernatants of diabetic fibroblasts tended to be reduced (albeit non-significantly) as compared with the supernatants of non-diabetic fibroblasts (diabetic: 441 ± 107 ng/mL, non-diabetic: 529 ± 85 ng/mL *p* = 0.14, Figure 2C). In the literature, in whole genome expression analyses of diabetic and healthy full skin, the procollagen C-endopeptidase enhancer 2 (pcolce2) that supports the cleavage of procollagen types I and II, was identified as being significantly downregulated in diabetics [41,42]. Therefore, qPCR was performed (Appendix A). Consistently, a trend of reduced mRNA expression of procollagen C-endopeptidase enhancer 2 (pcolce2) in diabetic subjects (Appendix A) was observed. These observations suggest that collagen synthesis was reduced in diabetic fibroblasts, as well. However, the data on secretion of MIF and PICP exhibited relatively large values of the standard deviation and *p*-values higher than 0.05 (Figure 2). Large values of the standard deviation and *p*-values higher than 0.05 seem to be associated with the usage of primary cells, e.g., the secretion of several pro-inflammatory factors from keratinocytes of diabetic donors tended to be increased (albeit non-significantly) [43,44]. A higher number of donors (if available) might help to find statistical significance. Increased secretion of IL-6 and the tendency to increase MIF and to decrease PICP secretion seem to be associated with diabetic fibroblasts.

### 3.4. Effects of MGO Treatment on the Secretion of IL-6, Macrophage Migration Inhibitory Factor (MIF) and Procollagen Type I C-Peptide in Dermal Fibroblasts

To investigate if the presence of MGO is sufficient for increased IL-6 secretion, dermal fibroblasts were treated with MGO. Remarkably, MGO concentration-dependently increased IL6 secretion in fibroblasts from non-diabetic donors (Figure 3A). Furthermore, MGO also induced increased IL-6 secretion in fibroblasts from diabetic donors (Figure 3B; non-diab.: 170 ± 49 pg/mL vs. 277 ± 131 pg/mL; diab: 179 ± 93 pg/mL vs. 256 ± 54 pg/mL). MGO treatment also resulted in significantly increased MIF secretion in both diabetic and non-diabetic fibroblasts (Figure 3C; non-diab.: 153 ± 47 ng/mL vs. 455 ± 219 ng/mL; diab.:172 ± 41 ng/mL vs. 403 ± 44 ng/mL). MGO, thus, is sufficient for triggering IL-6 and MIF secretion independently of the diabetic background.

Furthermore, MGO treatment resulted in reduced PICP production in fibroblasts from both diabetic and non-diabetic donors (diabetic: p_MGO_ = 0.028, non-diabetic: p_MGO_ = 0.011, Figure 3D; non-diabetic: 651 ± 204 ng/mL vs. 411 ± 140 ng/mL, diabetic: 520 ±145 ng/mL vs. 297 ± 219 ng/mL).

MGO treatment has been reported to result in increased ROS levels in human aortic endothelial cells [45]. Upon MGO treatment, the ROS levels were increased in fibroblasts from both diabetic and non-diabetic donors (Appendix A, non-diabetic: 32,146 ± 3391 a.u. vs. 34,657 ± 3312 a.u.; diabetic: 37,532 ± 3742 vs. 39,614 ± 5076 a.u.), which is consistent with former findings [46,47].

In sum, MGO treatment strongly increased the secretion of IL-6 and MIF and decreased the secretion of PICP in both diabetic and non-diabetic fibroblasts. The extent of MGO-induced secretion of IL-6 and MIF and reduced PICP secretion from non-diabetic and diabetic fibroblasts was clearly greater as compared with the extent of increased inherent secretion observed in fibroblasts from diabetic donors.

### 3.5. MIF Treatment Results in Reduced Glyoxalase I Activity in Diabetic Fibroblasts

Increased MGO levels in diabetic individuals are regarded to depend on the presence of high concentrations of the MGO precursor glucose and on down-regulation of the MGO-detoxifying glyoxalase system [48,49]. To discriminate if downregulation of glyoxalase I is due to decreased expression and/or decreased enzymatic activity of glyoxalase I, the expression of glyoxalase I protein was analysed using ELISA. Diabetic and non-diabetic fibroblasts expressed similar levels of glyoxalase I (463 ± 164 ng/mg total protein vs. 475 ± 145 ng/mg total protein) (Figure 4A). Next, glyoxalase I activity was analysed in terms of formation of S-d-lactoylglutathione. Remarkably, the glyoxalase I activity of diabetic fibroblasts (13.7 ± 2 U/min/mg) was decreased as compared with non-diabetic fibroblasts (11 ± 2 U/min/mg, *p* = 0.026, Figure 4B).

Decreased glyoxalase I activity might result in the elevated formation of MGO, which in turns results in the formation of AGEs [50]. The major MGO-derived AGE is 5-hydro-5-methylimidazolones (MG-H1) [25]. In this regard, AGE formation was estimated exploiting an MG-H1 (methylglyoxal 5-hydro-5-methylimidazolones) ELISA. The formation of MG-H1 tended to be increased (albeit not significantly) in fibroblasts from diabetic donors as compared with non-diabetic donors (Figure 4C; 121 ± 38 ng/mg total protein vs. 145 ± 38 ng/mg total protein)). Decreased glyoxalase I activity was not due to decreased glyoxalase I expression, leading to the hypothesis that a not yet identified factor negatively regulates glyoxalase I activity.

MIF has been suggested to aggravate diabetic neuropathy by suppressing glyoxalase-I [27]. In this regard, dermal fibroblasts were pre-treated with MIF for 24 h and GLO1 activity was analysed (Figure 4D). Remarkably, MIF specifically suppressed glyoxalase I activity in non-diabetic fibroblasts (Figure 4D; 9 ± 1 U/min/mg vs. 5 ± 2 U/min/mg). In diabetic fibroblasts, (already decreased intrinsic) GLO1 activity was as not responsive to MIF treatment (Figure 4D; 7 ± 2 U/min/mg vs. 6 ± 2 U/min/mg). The observation that MIF negatively regulates glyoxalase I activity is particularly important: MGO increases MIF secretion, which in turns inhibits the MGO-detoxifying GLO1, resulting in amplified MGO production. 

Comparable to GLO1, MIF suppressed PICP secretion in non-diabetic fibroblasts (Figure 3D; 651 ± 204 ng/mL vs. 411 ± 140 ng/mL). In diabetic fibroblasts, the (already decreased intrinsic) PICP secretion was not as responsive to MIF treatment (Figure 3D; 520 ± 145 ng/mL vs. 433 ± 142 ng/mL; non-diabetic: p_MIF_ = 0.015, diabetic: p_MIF_ = 0.074). These observations highlight MIF as a molecular link between hyperglycaemia and the changed regulation of cellular responses.

## 4. Discussion

In this study, the properties of dermal fibroblasts from diabetic and non-diabetic donors were differential analysed with regard to the secretion of IL-6, MIF, and PICP. In diabetic fibroblasts, the secretion of IL-6 was increased as compared to non-diabetic fibroblasts (Figure 2). Furthermore, a tendency of diabetic fibroblasts to increased MIF secretion and to decreased PICP secretion was observed (Figure 2). The latter data are not underlined by significance, yet relevance is given because they are pheno-copied upon treatment of primary dermal fibroblasts with MGO for 24 h: MGO increased secretion of IL-6 and MIF and decreased secretion of PICP (Figure 3). The extent of MGO-induced secretion of IL-6 and MIF was thereby greater as compared with the inherent secretion of IL-6 and MIF observed in fibroblasts from diabetic donors (Figure 2 and Figure 3). The latter observation resembles former observations showing that MGO treatment provokes the secretion of an array of immunomodulatory responses (including IL-6) in many experimental settings [14,18,19]. In conclusion, the exposure of skin cells to the glucose metabolite MGO seems to be an important trigger of immunomodulatory responses in the skin of diabetic individuals. Our observations reinforce the current paradigm that MGO links hyperglycaemia and increased immunomodulatory signalling in T2D patients.

Besides immunomodulatory responses, MGO exposure results in a reduction in PICP secretion from non-diabetic fibroblasts and diabetic fibroblasts (Figure 3D). PICP secretion tended to be (albeit non-significantly) reduced in diabetic dermal fibroblasts (Figure 2C). PICP production was reduced upon treatment of dermal fibroblasts with MIF (Figure 3D), showing that MIF is sufficient for inducing reduced PICP production. The pathway linking MIF and PICP production might involve MIF binding to CD74, which subsequently results in extracellular signal-regulated kinases (ERK)-dependent activation of the collagen I promotor ([51,52]. Secretion of MIF in turn was increased upon MGO treatment (Figure 3C). These observations suggest that MGO-induced reduction in the PICP production depends on MIF and highlight a new molecular link between hyperglycaemia and changed collagen production in human dermal fibroblasts. Reduced collagen production is most likely to (at least) contribute to the reduced wound healing observed in many T2D patients.

The changed expression of components of the collagen system could lead to changes in the structure of the ECM and the epidermis which, in turn, leads to a diminished protection of the skin against external influences such as bacterial wound infections and promotes the development of healing disorders and skin diseases.

In diabetic individuals, increased levels of MGO have been associated with decreased levels and/or decreased enzymatic activity of the MGO-detoxifying glyoxase system [19]. The observations of this study show that the expression levels of glyoxalase I were comparable in fibroblasts from diabetic and non-diabetic donors (Figure 4A). Remarkably, the activity of glyoxalase I was reduced in diabetic fibroblasts as compared with non-diabetic fibroblasts (Figure 4B). Comparable observations have been obtained from human keratinocytes, with the expression of glyoxalase I being comparable in keratinocytes from diabetic and non-diabetic donors but with the activity being reduced in keratinocytes from diabetic donors [43]. In this study, the activity of glyoxalase I was found to be reduced upon treatment of dermal fibroblasts with MIF(Figure 4D), suggesting that MIF activates a signalling pathway that negatively regulates glyoxalase I activity. This implies an additional attribute to MIF itself and to MIF affecting T2D [6]. A link between MIF and glyoxalase I is not entirely new, as MIF (in combination with MGO) treatment has been shown to reduce glyoxalase I protein expression [26]. Interestingly, glyoxalase I activity in diabetic fibroblasts was not responsive to MIF treatment (Figure 4D). The missing responsiveness of diabetic fibroblasts to MIF might be based on downregulation of the MIF receptor CD74 or on changed regulation of the MIF downstream signalling pathways including the PI3K-Akt, NF-κB, and AMP-activated protein kinase (AMPK) pathways [51,52].

The observations of this study lead to a model in which the immunomodulatory cytokine MIF exhibits pleiotropic effects, as it regulates metabolism, ECM formation, and inflammation. In the hyperglycaemic cell, an increased level of MGO is formed, as the level of its precursor glucose is elevated. MGO provokes increased secretion of MIF, which in turn blocks the MGO-degrading glyoxalase I, thus resulting in a self-accelerating cycle of MGO formation. Furthermore, MIF negatively regulates collagen production and promotes inflammation (Figure 5).

MIF appears to have a direct role in the progression of T2D by inhibiting glyoxalase I, leading to increased levels of MGO. Both inhibit collagen production, potentially resulting in deregulated wound healing. This leads to aberrant cell behaviour that is maintained in vitro. Hyperglycemia, one of the hallmarks of T2D, leads to increased formation of MGO. MGO then induces production of MIF and other inflammatory cytokines, leading to a state of inflammation (Figure 2 and Figure 3). MIF was found to decrease GLO1, which is part of the main system that detoxifies MGO (>90%), further urging on the elevated levels of MGO (Figure 4). Moreover, MIF as well as MGO reduced procollagen cleavage and, through that, active collagen production which, in turn, affected wound healing (Figure 3). In total, the findings of this study imply a connection between impaired glucose metabolism and inflammation as well as wound healing. Moreover, it implies a self-accelerating cycle in which MGO is perpetually formed and elevated.

These findings, for the first time to our knowledge, show a connection of dysregulated glucose metabolism and interfered ECM construction. Taken together, the diabetic phenotype of dermal fibroblasts (increased secretion of IL-6 and MIF and decreased PICP production) was phenocopied upon treatment of dermal fibroblasts with MGO. These observations strongly suggest that MGO acts as a key mediator of the skin pathology in diabetic individuals.

## Figures and Tables

**Figure 1 cells-10-02931-f001:**
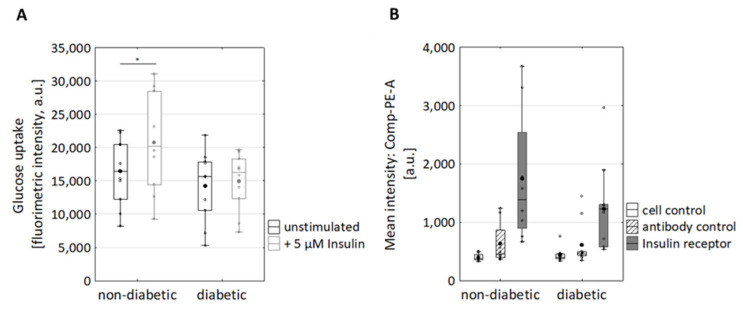
Impact of T2D on glucose uptake and Insulin receptor on the surface of dermal fibroblasts. (**A**) Glucose uptake assay. Fibroblasts from diabetic and non-diabetic donors were grown in starvation medium (DMEM without glucose and FBS) and treated with five µM insulin. Glucose uptake was analysed in terms of fluorescently labelled 2-(*N*-(7-nitrobenz-2-oxa-1,3-diazol-4-yl) amino) -2-deoxyglucose (2-NBDG; 100 μg/mL). * *p*  <  0.05, statistical significance was determined by Mann–Whitney-U-Test between groups and by Wilcoxon text for comparison between untreated and treated subjects; n_non-diabetic_ = 10; n_diabetic_ = 10. (**B**) Expression of the Insulin receptor on the cell surface of diabetic and non-diabetic fibroblasts was analysed using FACS. The cells in the absence of antibodies (=cell control) and cells with isotype antibodies (antibody control) served as controls, antibody against InsR n_non-diabetic_ = 8; n_diabetic_ = 10.

**Figure 2 cells-10-02931-f002:**
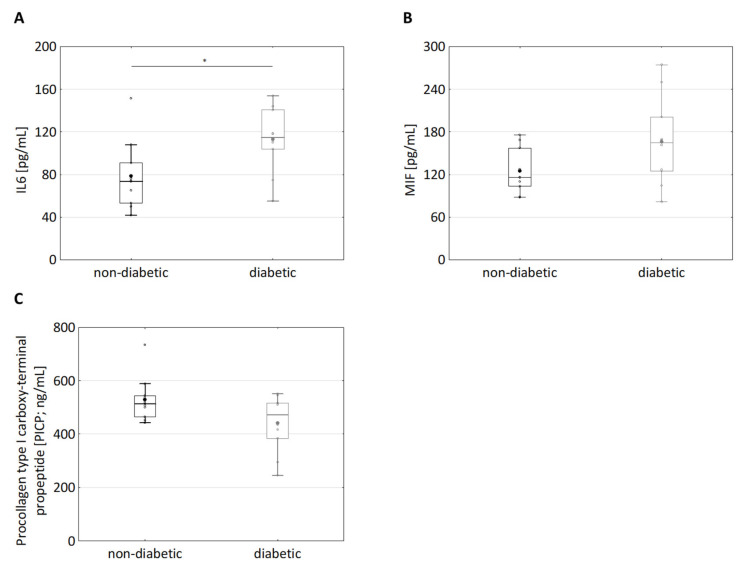
Difference in secretion of IL-6, macrophage migration inhibitory factor and Procollagen Type I C-Peptide in dermal fibroblasts from diabetic and non-diabetic donors. (**A**) The supernatants of cultured fibroblasts from diabetic and non-diabetic donors were analysed for IL6 levels using Homogeneous Time Resolved Fluorescence assay; n_non-diabetic_ = 10; n_diabetic_ = 10, * *p*  <  0.05. (**B**) The supernatants of cultured fibroblasts from diabetic and non-diabetic donors were analysed for MIF using ELISA; n_non-diabetic_ = 10; n_diabetic_ = 10. (**C**) The supernatants of cultured fibroblasts from diabetic and non-diabetic donors were analysed for procollagen type I C-peptide using ELISA. Statistical significance was determined by Mann–Whitney-U-Test between groups; n_non-diabetic_ = 10; n_diabetic_ = 10.

**Figure 3 cells-10-02931-f003:**
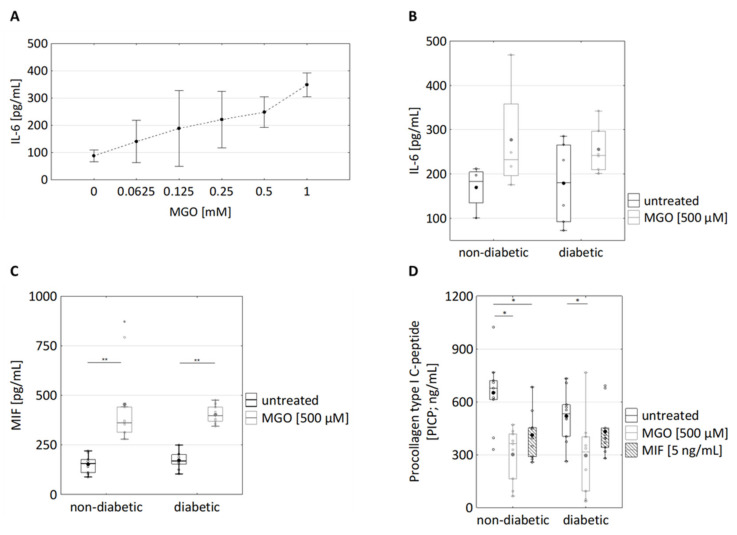
Effects of MGO and MIF treatment of dermal fibroblasts on the secretion of immunomodulatory cytokines and procollagen type I C-peptide. (**A**) Cultured dermal fibroblasts were treated with the indicated MGO concentrations (ranging from 0.0625 to 1 mM) for 24 h. The supernatants were analysed for IL6 levels using Homogeneous Time Resolved Fluorescence assay n = 6 (3 non-diabetic, 3 diabetic). (**B**) Cultured fibroblasts from diabetic and non-diabetic donors were treated with 500 µM MGO for 24 h. The supernatants were analysed for IL6 levels using Homogeneous Time Resolved Fluorescence assay; n_non-diabetic_ = 5; n_diabetic_ = 7. (**C**) The supernatants of cultured fibroblasts from diabetic and non-diabetic donors were analysed for MIF using ELISA. They were treated with 500 µM MGO for 24 h; n_non-diabetic_ = 9; n_diabetic_ = 10. (**D**) The supernatants were analysed for procollagen type I C-peptide using ELISA. Fibroblasts from diabetic and non-diabetic donors were treated with 500 µM MGO and 10 ng/mL MIF for 24 h; n_non-diabetic_ = 9; n_diabetic_ = 10. Statistical significance was determined by Mann–Whitney-U-Test between groups and by Wilcoxon test for comparison between untreated and treated; * *p*  <  0.05, ** *p*  <  0.01.

**Figure 4 cells-10-02931-f004:**
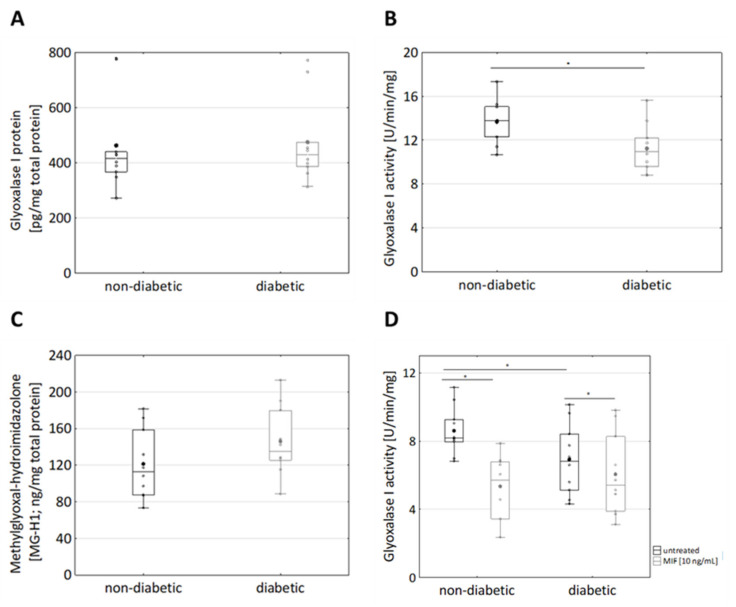
Glyoxalase I system in non-diabetic and diabetic dermal fibroblasts. Cultured fibroblasts from diabetic and non-diabetic donors were lysed by adding PBS and 3 freeze/thaw cycles. (**A**) The expression of glyoxalase I protein was measured via ELISA; n_non-diabetic_ = 10; n_diabetic_ = 10. (**B**) Glyoxalase I activity was analysed in terms of formation of S-D-lactoylglutathione, which was photometrically measured as a change in absorbance at 240 nm. Statistical significance was determined by Mann–Whitney-U-Test between groups and by Wilcoxon test for comparison between untreated and treated, * *p* < 0.05, n_non-diabetic_ = 10; n_diabetic_ = 10. (**C**) 5-hydro-5-methylimidazolone (MG-H1) was detected in lysates of diabetic and non-diabetic fibroblasts. (**D**) Fibroblasts from diabetic and non-diabetic donors were pre-treated with 10 ng/mL MIF for 24 h and lysed by adding PBS and three freeze/thaw cycles. Glyoxalase I activity was analysed in terms of formation of S-D-lactoylglutathione, which was photometrically measured as a change in absorbance at 240 nm. n_non-diabetic_ = 10; n_diabetic_ = 10.

**Figure 5 cells-10-02931-f005:**
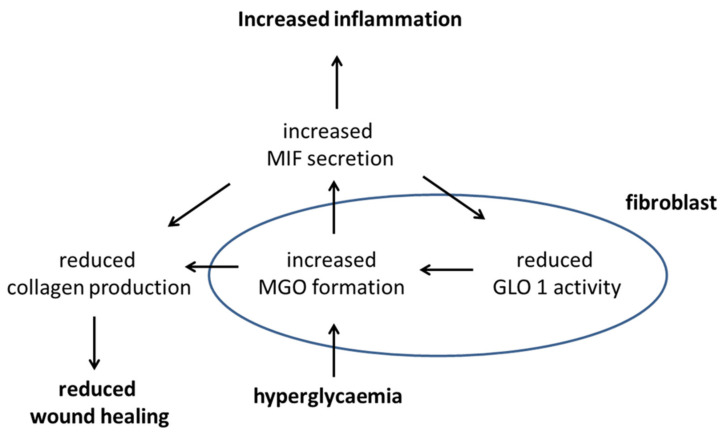
Putative signalling pathway under hyperglycaemic conditions on how hyperglycemia affects wound healing and inflammation.

**Table 1 cells-10-02931-t001:** Demographic data of the 10 donors with type 2 diabetes and the 10 non-diabetic donors. Values are expressed as mean (±SD).

	Group 1: Donors with T2D	Group 2: Donors without T2D
n total n female n male	10 3 7	10 3 7
Age (y)	54 (±9)	54 (±9)
HbA_1c_ (%)	7.06 (54 mmol/mol) (±1.03)	5.44 (36 mmol/mol) (±0.22)

## Data Availability

All relevant data are within the paper and its Appendix A.

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
