# Peer review of "Evaluation of Immunomodulatory Responses and Changed Wound Healing in Type 2 Diabetes—A Study Exploiting Dermal Fibroblasts from Diabetic and Non-Diabetic Human Donors"

_cells, 2021, doi:10.3390/cells10112931_

Round 1

Reviewer 1 Report

The manuscript is of significant interest and largely well written. The presentation of results of should be improved.

Figure 1A: Please indicate what was compared and where the difference was significant. Where is the asterisk?

Throughout the manuscript, results of measurement are reported with inappropriate “accuracy”. The number of decimal places is not acceptable. Example in line 388: “32145.96±3391.39 vs. 37532.40±3742.20“. If you are not sure that your error of measurement is smaller than 0.001%, please change these numbers in the text.

Lines 378-383: This paragraph does not present results but discuss the data. Only short interpretations should be included in the Results section.

Line 524-528: Please rephrase the concluding sentences. The message should be clearer. The statement that the skin is the biggest organ and therefore very important is not convincing and should be deleted.

Reviewer 2 Report

It is interesting to study on dermal fibroblasts from diabetic and non-diabetic human donors. The research was properly planned. The observations suggesting that collagen synthesis was reduced in diabetic fibroblasts confirms previously published studies. MGO treatment resulted in significantly increased MIF secretion in both diabetic and non-diabetic fibroblasts. MGO links hyperglycaemia and increased pro-inflammatory signalling in T2D patients. Authors strongly suggest that MGO acts as a key mediator of the skin pathology. However:

  1. The presented results relate to a small study group, compared to other publications (even cited by Authors), this number was even lower in some trials, e.g. 3 non-diabetic, 3 diabetic or n non-diabetic=5; n diabetic=7.

Additionally,

  1. All results were presented as mean ± standard error of the mean (S.E.M.), what’s correct. However significant/large values of the standard deviation (even > 40%, e.g. 455,17 ± 219,16 or 277,4 ± 131,07 pg/mL) and p-values higher than 0.05 make the obtained results questionable.

It is difficult to conclude whether such values are due to inaccuracies or from a relatively small number of patients, it should be commented.

  1. There is no need to specify the color in the caption, e.g. (grey) or (green), when the publication is black and white.
  2. List of references is negligent. Authors should apply Reference Formatting;
  • complete references with doi or year, volume, issue, page numbers, e.g. reference 48, 47, 41;
  • or appropriate journal title, e.g. line 596 reference 27 sholud be J Diabets Res.
  1. Should be period at the end of the sentence, e.g. line 501, 160 or 175.

Reviewer 3 Report

In this study, Nickel et al., isolate fibroblasts form diabetic patients and non-diabetic controls to examine differences in their immunomodulatory responses. The study shows that diabetic fibroblasts are less efficient in glucose uptake in response to insulin, have increased IL-6 production and decreased procollagen type I C peptide (PCIP) synthesis. Interestingly, PCIP   production was decreased by both the glucose metabolite methylglyoxal (MGO) and macrophage migration inhibitory factor (MIF) in non-diabetic fibroblasts and by MGO in diabetic fibroblasts. Glyoxalase activity, enzyme that degrades MGO, is reduced in fibroblasts isolated from diabetic patients compared to those isolated from non-diabetic patients. Interestingly, treatment of both diabetic and non-diabetic fibroblasts with MIF results in decreased activity of glyoxalase.

 Furthermore, MGO promotes IL6 and MIF production in both diabetic and non-diabetic fibroblasts.

This is a nice study that aims to examine mechanisms impairing wound healing in diabetic patients.

My main concerns are

  • in many instances, results shown are not statistically significantly different between the groups in many of the results shown and some of the points are not clearly made. Although it is understandable that when using primary human cells, the variability may be high it is also difficult to draw conclusions based on statistically non-significant values.
  • Many figures show the effect of MGO in fibroblasts independent of whether these are isolated from diabetic or non-diabetic donors. These findings of course have merit. However, they are not representative of inherent differences between diabetic and non-diabetic fibroblasts. The authors may want to reorganize their manuscript to highlight these inherent differences between diabetic and non-diabetic fibroblast and then to show the effect of MGO. Additionally, although in certain figures it is shown that IL6 and PCIP production are different in the diabetic and non-diabetic fibroblasts in other figures this is not the same (see below)

Figure 1A: addition of representative flow plots would be beneficial. Also, cells with isotype antibody controls should have been included to confirm specificity

Figure 2B:  differences between diabetic and non-diabetic fibroblast MIF production are not statistically significant

Figure 2C: differences between diabetic and non-diabetic fibroblasts PICP production are not statistically significant; the same for mRNA levels od C endopeptidase enhancer 2 (shown in figure S2). However, in figure 3D is it shown that there is a sizable difference of PCIP between diabetic and non-diabetic fibroblasts in control conditions, although the p value is not reported.

In addition, in some of their plots there is slight disagreement; for example, figure 2A shows increases IL-6 production in diabetic and non-diabetic fibroblasts. Figure 3B shows IL-6 production from diabetic and non-diabetic fibroblasts in the presence or absence of MGO. In this figure there is no observable difference of IL6 production between diabetic and non-diabetic fibroblasts in the non-stimulating condition. This of course may be due to the fact that there was a lower n used (n=3); however, if possible, it would be good to repeat this experiment with the appropriate n to reproduce the result in figure 2A. Addition of extra ns may also result in differences in IL-6 production in response to MGO.

The same concern between figure 4B and 4D: can the authors comment whether the differences between nondiabetic and diabetic glyoxalase activity are statistically significant as shown in figure 4B?

Similar concerns in figure 2C and 3D as mentioned above.

On the other hand, the authors show differences in ROS production in non-diabetic vs the diabetic fibroblasts in their supplemental figures. Interestingly this is the only parameter shown that the differences between the group remain statistically significant even in the presence of MGO or MIF. In lines 391-392 the authors mention: “This implies that diabetic cells were more stressed, conforming data found in literature.” Please cite the literature

 In lines 391-392 referring to MGO treatment the authors mention: “However, this did not affect proliferation of the cell that were assessed by nuclei counting as well as BrdU assay.”  Supplemental figure 3B shows statistically significant decrease in BrdU incorporation in the presence of MGO. Please clarify. 

Lines 439-444 describe figure 3D and not 4D

Round 2

Reviewer 3 Report

I am still confused on what exactly is inherent to the diabetic fibroblasts and what is only in response to MGO.

The data shown so far only IL-6 (in some instance), glo 1 and glucose uptake are shown different in diabetic fibroblasts; the rest data show the fibroblast response to MGO independently of diabetic status.

Thus, the observed results could be the result of mainly MGO; and if MGO is higher in the skin of diabetic patients this could be the reason for increased inflammation and lower PCIP production; would this be an accurate statement? If yes, I would suggest that the authors reorganize their discussion and result conclusion to reflect this.

I sympathize with the authors’ statement that primary cells can be the source of high variability.
Lack of statistical significance leads to low confidence of the results presented and does not support a conclusion of up- or down- regulation .

I am not sure that I understand what the “wrong set up of experiments” mean.

Author Response

Dear reviewer,

Kind regards,

Kimberly Nickel
